# Role of invasive *Aedes (Aedimorphus) vittatus* in chikungunya virus transmission in the newly-invaded coastal island city of Mombasa, Kenya

Francis M. Musili[1,2]*, James Mutisya[1], Solomon Langat[1], Betty Chelangat[1], Victor Anyango[3], Edith Chepkorir[1], Samson Konongoi[1], Rosemary Sang[1], Armanda Bastos[2,4], Joel Lutomiah[1]

**1** Center for Virus Research, Kenya Medical Research Institute, Nairobi, Kenya, **2** Department of Zoology & Entomology, University of Pretoria, Pretoria, South Africa, **3** Department of Biology, Texas Tech University, Lubbock, Texas, United States of America, **4** Department of Veterinary Tropical Diseases, University of Pretoria, Onderstepoort, South Africa

\* fmusili85@gmail.com

## Abstract

### Background

Countrywide routine entomological surveillance studies in Kenya from 2006 first identified *Aedes vittatus,* in small numbers (4 specimens), in Mombasa city in 2014 during a dengue outbreak investigation. Significant numbers (1,648 specimens) were collected in January 2018 during a chikungunya outbreak investigation. The presence of *Ae. vittatus,* and a competent vector of chikungunya virus, complicates disease epidemiology and control efforts in Mombasa underscoring the need to determine its bionomic factors.

### Methods

In June 2021 and December 2021, we conducted mosquito sampling at multiple sites in Mombasa, an island city county, using $CO_2$-baited Biogent sentinel (BGS) traps and human landing collection (HLC) methods. The collected mosquitoes were identified morphologically, and *Ae. vittatus* species were confirmed further by molecular characterization based on the cytochrome c oxidase I gene (*cox1*). Virus isolation from mosquitoes pools was performed in Vero ccl-81 cell cultures. Cultures showing cytopathic effects were harvested and genome-sequenced using the Illumina MiSeq platform to identify the infecting virus.

### Results

A total of 11,435 mosquitoes were collected; 7,250 by BGS traps and 4,185 by HLC. Overall, *Ae. aegypti* was the dominant species, accounting for 32.6% (n = 3,725), followed by *Culex quinquefasciatus* which accounted for 31.8% (n = 3,638) and

**Data availability statement:** The dataset generated and analyzed in this study are provided within the manuscript and its supplementary materials. COI sequences for Aedes vittatus are available in NCBI under accession numbers PX499745–PX499753. The Chikungunya isolate sequence data is available in the GISAID database under accession number EPI_ISL_20252512.

**Funding:** This study was funded by the Kenya government through the National Research Fund (NRF) Kenya, Grant number NRF/1/MMC/28 awarded to JL. The funders had no role in study design, data collection and analysis, decision to publish, or preparation of the manuscript.

**Competing interests:** The authors have declared that no competing interests exist.

*Ae. vittatus* at 24.4% (n = 2,789). *Aedes aegypti* (n = 2,216; 43%) predominated in HLC collections, followed by *Ae. vittatus* (n = 1,598; 31%). Although mosquito biting rates per person per hour (b/p/h) were higher for *Ae. aegypti* (3.2 b/p/h) than *Ae. vittatus* (2.4 b/p/h), the difference was not statistically significant (t = 6.0081, df = 1, p-value = 0.105). Chikungunya virus isolate belonging to the East Central South African genotype was isolated from a pool of female *Ae. vittatus*.

## Conclusion

*Aedes vittatus* was found widely distributed across the island city in significant numbers, suggesting that the species, which is predominant to most rural areas in Kenya, has invaded the city and successfully established. The presence of this species in the city, a confirmed vector of CHIKV, and along with *Ae. aegypti* which is the principle vector of CHIKV, potentially magnifies the risk of chikungunya outbreaks. This highlights the significance and need for integrated vector control strategies.

## Author summary

*Aedes vittatus* poses a growing public health threat due to its extensive geographic range and proven capacity to transmit various medically important arboviruses, such as yellow fever, dengue, chikungunya, and Zika viruses. Mombasa city is becoming endemic for most emerging arboviruses, such as chikungunya and dengue, which are transmitted by the widely distributed *Aedes* mosquitoes in the area. The *Ae. vittatus* adaptation to diverse natural and artificial breeding sites in the city increases its ability to survive and proliferate. Due to its invasive capacity, enhanced by climate change, global trade and transportation, *Ae. vittatus* poses an increasing epidemiological threat in the city. We report the high abundance of *Ae. vittatus*, its high proclivity to humans and detection of chikungunya virus in Mombasa city. Its presence in multiple locations on the island suggests that it has quickly adapted to the urban environment which may enhance the arboviral outbreaks already reported in the region. Adaptability of this mosquito to human-inhabited areas increases the risk of pathogen transmission to humans, underscoring the urgent need for enhanced surveillance by the vector control program at the Ministry of Health for timely interventions.

## Introduction

Chikungunya virus (CHIKV) is an RNA virus in the *Alphavirus* genus that is primarily transmitted by *Aedes* mosquitoes such as *Ae. aegypti* and *Ae. albopictus.* It remains a significant global public health concern [1]. In Kenya, genetic analysis of CHIKV during the recent outbreaks in Mombasa between 2017 and 2018 revealed key viral mutations, including the E1:V80A mutation in an emerging strain, which may have affected the virus's transmissibility and virulence [2]. Genome sequencing data

provided valuable insights into the virus's evolutionary trends and its potential adaptations to mosquito vectors [2,3]. Entomological investigations during the outbreaks pointed to the probable role of *Cx. quinquefasciatus* mosquitoes in the transmission of CHIKV in Mombasa, suggesting that while *Aedes* species remain the primary vectors, *Culex* populations might also play a role in spreading the virus [4]. Mombasa city is experiencing upsurge of dengue and chikungunya outbreaks, during dengue outbreak the positive cases range between 60–72% compared to 30–52% during non-outbreak periods [5] while chikungunya seroprevalence during outbreaks ranges from 50-65% [6].

Invasive arboviral vectors and mosquito-borne diseases of public health and veterinary importance have steadily increased globally [7]. Invasive mosquito species (IMS) are a major concern due to their ability to invade new areas, establish populations, and transmit human viral diseases [8]. Additionally, they alter the ecosystems and resource competition, which may result in the displacement of existing species [9]. As IMS, particularly *Aedes* mosquitoes, expand their ranges due to climate change and human activities, mosquito-borne diseases such as dengue, chikungunya, yellow fever, and Zika are also introduced to new regions [8,10]. Most of these vectors are highly anthropophilic, feeding preferentially on humans, which links them to global disease outbreaks [10]. This highlights the need for improved surveillance to enable public health professionals, researchers, and mosquito control organizations to implement new strategies to manage IMS and reduce the risk of mosquito-borne diseases. Early detection and effective elimination programs are key to effective IMS eradication efforts [11].

*Aedes vittatus* is one of the globally invasive species, with a broad geographic range, in tropical and subtropical regions of Asia, Africa, and the Mediterranean area of Europe, it is commonly found as either a sylvatic or peridomestic mosquito, particularly in rural areas [12,13]. The species was first placed under subgenus *Stegomyia* due to morphological similarities; but, subsequently placed under subgenus *Aedimorphus* and later on under the subgenus *Fredwardsius.* This new sub genus has not been well recognized hence the vector remains in subgenus *Aedimorphus* as described by Huang in 1977 [14]. It is predominantly a rock- pool breeder in Africa, though it can breed in diverse macro- and microhabitats [6]. The species is both a diurnal and nocturnal feeder, biting humans and animals, and plays a significant role in maintaining and transmitting several arboviruses [15]. However, its aggressive biting behavior, may contribute to the spread of various pathogens in the African region, such as CHIKV, dengue virus (DENV), yellow fever virus (YFV), and Zika virus (ZIKV) [16–18]. Additionally, the presence of the vector in urban environments may facilitate the ongoing circulation of CHIKV [19]. The widespread *Ae. vittatus* presence in urban, peri-domestic, and rural environments in Kenya, along with their associated *Ae. aegypti* indices, has raised concerns about the potential for both urban and rural transmission of arboviruses [13,20,21].

*Aedes vittatus* is a demonstrated competent vector of CHIKV in Kenya, showing effective viral replication and dissemination after experimental viral exposure [22]. Previously, during the routine surveillance in 2013–2014, there was a very low abundance of the species in Mombasa city [20]. However, subsequent entomological surveillance during chikungunya outbreak in 2018 in the region has shown a significant increase in its abundance, as evidenced by both larval and adult collections [4]. The growing presence of *Ae. vittatus* in coastal Kenya is a significant public health concern due to its potential role in the transmission of arboviral infections in the region. Its role in transmission and historical prevalence in areas affected by CHIKV highlights the need for targeted vector control strategies.

Mombasa is an important port city in the East and Central Africa region, and a global tourist destination, making it vulnerable to the introduction and distribution of invasive vectors and infectious agents [23]. The city has experienced a substantial increase in overall visitor arrivals and a high volume of cargo coming from both mainland and overseas [24]. The city has also become endemic to chikungunya and dengue with sporadic outbreaks being experienced [25]. The involvement of the invasive mosquito *Ae. vittatus* in the transmission of CHIKV in coastal Kenya is a major concern, especially in light of the expanding geographic spread of arboviral infections across East Africa. An entomological field surveillance was conducted in Mombasa islands in June to December 2021 to determine mosquito species abundance, diversity and attractiveness to humans, and virus isolation to determine the risk of arbovirus transmission in the region.

## Methods

### Ethics statement

The study was approved by the Kenya Medical Research Institute's (KEMRI) Scientific Ethics Review Unit (SERU 4999). Written, signed consent was obtained from all adults volunteering for HLC. The entire consenting process was overseen by the PI, supported by the county-level public health supervisor. All consenting participants underwent mandatory prophylactic vaccination against yellow fever virus funded by the project before being comprehensively trained in mosquito sampling procedures through demonstrations by the project's lead entomologist who was also the principal investigator (PI). Each volunteer received a copy of their signed consent form and another copy while another copy was kept in KEMRI. Verbal consent was obtained from the heads of households to allow the vector sampling team access to their residential compounds for the purpose of mosquito collection.

### Study area

The study sites were selected in the city of Mombasa in areas that have previously experienced most severe chikungunya outbreaks, including Mama Ngina Grounds (MG), Kenya Ports Authority (KPA), Mombasa Railway station (RA) and Majengo in Mvita (Fig 1). These sites were selected to capture potential variation in mosquito ecology across urban land-use types making it a key sentinel location for arbovirus surveillance. Mombasa, Kenya, has a hot tropical climate classified as wet and dry, shaped by seasonal monsoon winds. Temperatures remain high throughout the year, averaging 27.9°C. The city experiences two main rainy seasons: the long rains from April to June and the short rains between October and December. Its coastal location along the Indian Ocean contributes to consistently high humidity levels, averaging around 80%.

### Consenting, enrolment and training of HLC volunteers

The Human Landing Collection (HLC) method is considered as a gold standard for measuring exposure of humans to mosquito bites and for estimating the human biting rate [26]. Human Landing Collection (HLC) was used to collect adult *Aedes* mosquitoes and assess the degree of anthropophily among different populations. Eligible participants were adults (≥18 years old) who resided within the sampling areas. The consent process included a clear explanation of the purpose of the study, its potential risks and benefits and the need for HLC, emphasizing that participation was entirely voluntary and that participants could withdraw at any time without providing a reason. This was done in both English and Kiswahili, which is the national language in Kenya. Community meetings were held in all four sites within the county to introduce the research project and staff to the local government administrators and residents.

### Human landing collection process

This process involved clearly explaining the objectives of the study, the sampling procedures to be carried out, and addressing any questions or concerns raised by the household members before seeking their permission. Mosquitoes were collected outdoors at each study site during the long rainy seasons in June 2021 and short rainy season in December 2021 when mosquito population densities are highest. Sampling was conducted for six consecutive days during each season, targeting the active feeding times of *Aedes* mosquitoes species, which feeds during the day. Ten trained consenting participants were involved in the HLC process. Mosquitoes were collected each season for six days from 9:00 am to 12:00 am and 2:00 pm to 5:00 pm to cover the two peak feeding periods for *Aedes species*. During each collection period, participants wore long-sleeved shirts, leaving only their lower part of the leg exposed to ensure mosquitoes could only land on that specific area. When a mosquito landed, the participant swiftly and carefully captured it using a biting collection (BC) tube before it had the chance to bite. Once the mosquito entered the tube, it was immediately secured with a cotton wool stopper.

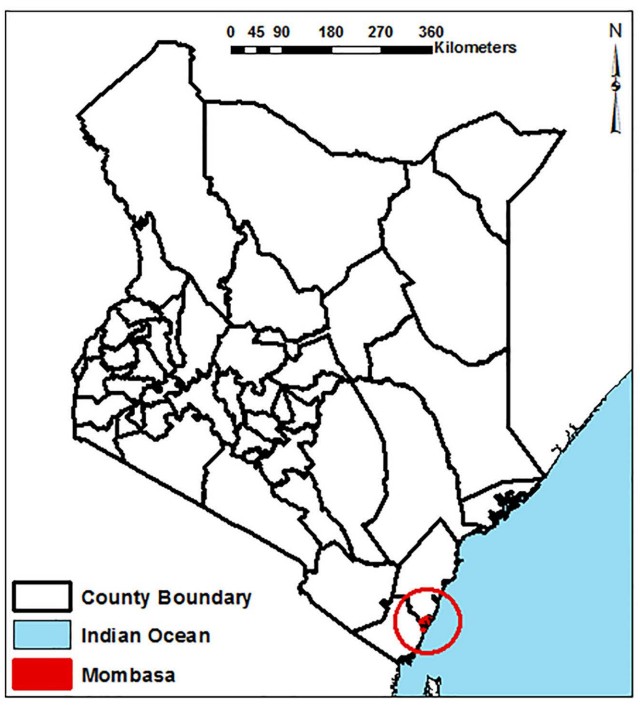
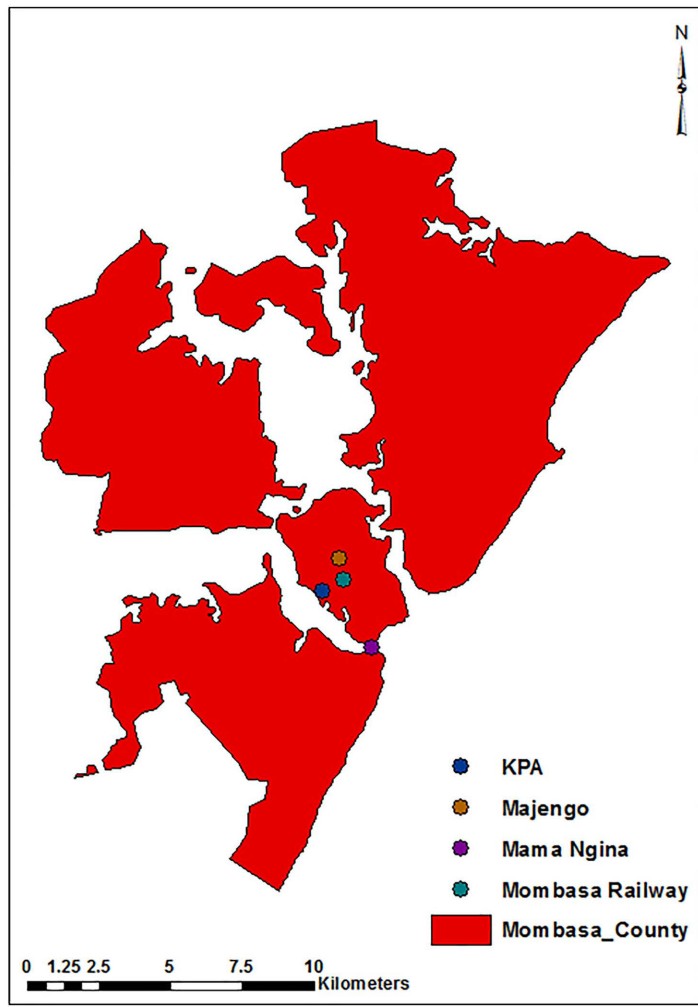

**Fig 1. Map of Kenya showing the entomological sampling sites in Mombasa city.** The map was designed using ArcMap 10.2.2 with the ocean and lakes base layer derived from Natural Earth, a free GIS data source (naturalearthdata). The locations were collected using a GPS gadget (garmin etrex 20, https://buy.garmin.com/en-US/US/p/518046), and the county boundaries for Kenya derived from Africa Open data (https://africaopendata.org/dataset/kenya-counties-shapefile).

Each BC tube was labeled with details including the date, time, location, and habitat. Human landing catches were performed for 3-hour sessions, with a 15-minute break each hour, during which time the collector would rotate to another collection point within the study area. The PI and county-level supervisor monitored the collection process to ensure adherence to the protocol. Mosquitoes were counted while still in the BC tubes labelled by area and date of collection. The mosquitoes were then anesthetized using trimethylamine for three minutes [27,28]. After anesthesia, the mosquitoes were preserved in a dry shipper and transported to the laboratory for species identification.

**Biogent sentinel (BGS) traps**

The BGS traps, regarded as the gold standard for adult *Ae. aegypti* surveillance, were deployed outside houses for 24 hours in the same locations where HLC was conducted for six days. Each trap was baited with 0.5 kg of dry ice ($CO_2$) placed in an insulated container (igloo) beside it as an attractant to maximize the collection. Mosquitoes were retrieved

from the traps twice daily, in the morning and evening, and transported to a temporary site laboratory. The collection bags containing the trapped mosquitoes were carefully packed into biohazard bags. The mosquitoes were anesthetized using trimethylamine for three minutes [27,28] and transported to the laboratory for identification.

## Mosquito identification

Collected mosquitoes were morphologically identified to species level following the classification keys of Edwards (1941) and Jupp (1996) [29,30], pooled (up to a maximum of 25 mosquitoes per pool) by species, sex, collection date and site, and preserved at -80°C for subsequent laboratory analysis. Representative *Ae. vittatus* were preserved individually for molecular confirmation of morphological identification.

## DNA extraction, PCR and sequencing for molecular identification

Morphologically identified *Ae. vittatus* specimens were homogenized in phosphate buffer sulphate using 6–10 zirconia beads at 4.0 m/s with the Omni Bead Ruptor-24. Following centrifugation, genomic DNA was extracted using Qiagen DNA extraction kits. The mitochondrial cox1 gene was targeted using universal primers: forward (LCO1490: 5′-GGTCAACAAATCATAAAGATATTGG-3′) and reverse (HCO2198: 5′-TAAACTTCAGGGTGACCAAAAAATCA-3′), designed to amplify a ~710 bp fragment [31]. PCR amplification was carried out in 25 µL reaction volumes containing 12.5 µL of 2X Taq PCR Master Mix (Qiagen), 1 µL of each primer (10 pmol/µL), 3 µL of template DNA, and 7.5 µL of nuclease-free water. The thermal cycling conditions included an initial denaturation at 94°C for 3 minutes; followed by 35 cycles of denaturation at 94°C for 30 seconds, annealing at 49°C for 30 seconds, and extension at 72°C for 1 minute; with a final extension at 72°C for 10 minutes. Amplified products were visualized on a 1.5% agarose gel stained with Diamond nucleic acid dye (Promega). PCR products were then purified using the Exo-CIP Rapid PCR Cleanup kit (BioLabs), with incubations at 37°C for 15 minutes and 85°C for 15 minutes, prior to Sanger sequencing. Raw cox1 sequences were quality-trimmed using Chromas v2.6.6 (https://technelysium.com.au/) to remove low-quality ends. The cleaned sequences were compared against GenBank entries using BLASTn, applying a cut-off e-value of 1e−50 for significant similarity. Sequence alignment was conducted using MUSCLE in MEGA7, and haplotype analysis was performed in DnaSP v6. Phylogenetic relationships were inferred using maximum likelihood (ML) trees constructed from the identified haplotypes and relevant reference sequences obtained from GenBank.

## Mosquito homogenization

Identified mosquitoes were pooled into groups of up to 25, resulting in a total of 843 pools. Each pool was homogenized with 2.0 mm zirconium beads for 40 seconds using a Mini-Beadruptor-16 (Biospec, Bartlesville, OK, USA) in 1000 µL of homogenization medium. The medium consisted of minimum essential medium (MEM) supplemented with 15% fetal bovine serum (FBS) (Gibco, Life Technologies, Grand Island, NY, USA), 2% L-glutamine (Sigma-Aldrich), and 2% antibiotic/antimycotic solution (Gibco, Life Technologies, Grand Island, NY, USA). Following homogenization, the samples were centrifuged at 10,000 rpm for 12 minutes at 4°C using an Eppendorf benchtop centrifuge (Eppendorf, USA). The supernatant was then collected and transferred to new tubes for subsequent use in cell culture.

## Virus isolation

Virus isolation was performed on all 843 pools in Vero cells (African green monkey kidney) (CCL-81™). Briefly, Vero cells were grown overnight at 37°C and 5% $CO_2$ in MEM supplemented with 2% L-glutamine, 2% penicillin/streptomycin/amphotericin, 10% fetal bovine serum (FBS), and 7.5% NaHCO3 in 24-well plates (Corning, Incorporated). At 80% confluence, 50 µL of the clarified supernatant from individual pools was inoculated into each well. The plates were incubated for 1 hour in a humidified incubator at 37°C and 5% $CO_2$ with gentle rocking of the plates every 15 minutes for virus adsorption. Following incubation, 1ml of maintenance medium, comprising of MEM supplemented with 2% glutamine,

2% penicillin/streptomycin/amphotericin, 2% FBS, and 7.5% NaHCO3, was added. The cultures were again incubated at 37°C and 5% $CO_2$ and monitored daily for cytopathic effects (CPE) for a maximum of 14 days. Cultures exhibiting CPE were harvested and further passaged by inoculating onto fresh monolayers of Vero cells (CCL 81™) in 25-cm3 cell culture flasks. After two successive passages, the supernatants of virus-infected Vero cell cultures exhibiting CPE of approximately 70% were harvested for virus identification through next-generation sequencing.

### Library preparation and next generation sequencing

Viral particles from cultures showing reproducible CPE were filtered through 0.22μm filters (Millipore, Merck). Viral RNA was extracted using the QIAamp Viral RNA Mini Kit (Qiagen, Hilden, Germany). Paired-end libraries for high-throughput sequencing were prepared using the Illumina RNA Prep, ligation kit following the manufacturers recommended protocol (Illumina, USA). Sequencing was carried out using MiSeq Reagent kit V3 (Illumina, USA), in a 300 bp paired-end cycle sequencing format.

### Sequence analysis and virus identification

Sequence analysis was carried out using CZ ID bioinformatics pipeline v8.3.15 (https://czid.org/). The pipeline is an open-source sequence analysis platform that takes raw sequence data as input and performs quality control, de-hosting, duplicate removal, as well as assembly and identification of viruses. Following detection, the consensus viral genome was obtained using CZ ID's consensus genome pipeline v3.5.0. The generated sequence was compared against GenBank nucleotide database using BLASTn, with a cut-off e-value of 1e−50. Non-host reads that mapped to CHIKV were aligned to Chikungunya virus isolate KLF_84548 (MT526800.1), a member of the ECSA genotype which was closest to our sequence by BLAST analysis. Chikungunya virus consensus genome with an average coverage depth of 13874.0x and coverage breadth of 99.1% was obtained.

### Phylogenetic analysis of the identified RNA viruses

Phylogenetic inference was carried out using Molecular Evolutionary Genetics Analysis v.12.0 (MEGA12) program [32]. Publicly available sequences belonging to the three major lineages of CHIKV; ECSA, Asian and West African were randomly selected and downloaded from Genbank. The downloaded sequences were combined with the sequence generated in the study. The combined set of sequences were aligned using Muscle software embedded in MEGA12 and edited using Bioedit program. Maximum likelihood phylogenetic analysis was carried out using MEGA12 software, based on the GTR+G+I model of nucleotide substitution as determined through model selection option in MEGA12. The analysis was run for 1,000 bootstrap estimates and the generated tree was visualized in Figtree v1.4.

### Data analysis

The mosquito collection data were cleaned and descriptive statistics such as means and standard deviation, were calculated. All statistical analyses were carried out in R (v4.3.1) software [33] and significance level was inferred at $p < 0.05$. Mosquito alpha diversity was determined for each sampling method, Scaling with Ranked Subsampling (SRS) in R version 3.3.1 [34]. Beta diversity analysis was performed to assess differences in mosquito community composition between collection methods. Bray-Curtis dissimilarity matrices were calculated from the SRS-normalized abundance data, and permutational multivariate analysis of variance (PERMANOVA) was conducted using the adonis2 [35] function in the vegan R package, with 999 permutations. Biting rates and mosquito abundance were compared using One Sample t-test and Kruskal-Wallis test respectively. All the data sets were analyzed at $\alpha = 0.05$ level of significance. The human biting rate (HBR) was determined using the formula below [36].

$$HBR = \frac{\text{Total number of collected mosquito of a particular species}}{\text{Total number of human baits per total hours of collection}}$$

## Results

### Mosquito species diversity and abundance

A total of 11,435 mosquitoes were collected from three distinct sites in Mombasa, Kenya: Kenya Ports Authority (KPA), Mombasa Railway Station RS, and Mama Ngina Drive (MN), during both the short and long rainy seasons, between June 2021 and December 2022. The collected mosquitoes belonged predominantly to the genera *Aedes* (60.0%; n = 6,862), *Culex* (38.9%; n = 4,443), and *Anopheles* (0.1%; n = 11), while other genera accounted for 1.0%; n = 121. Overall, the most abundant species was *Ae. aegypti* (n = 3,725; 32.6%), followed by *Cx. quinquefasciatus* (n = 3,638; 31.8%;) and the invasive *Ae. vittatus* (n = 2,989; 26.1%). For the HLC a total of 4,185 (36.6%) mosquitoes were collected of which *Ae. aegypti* (n = 2,216; 53%) is having the highest proclivity to the humans followed by *Ae. vittatus* (n = 1,438; 34.4%). The chances of finding *Ae. aegypti* mosquitoes by HLC were higher in the afternoon than in the morning although not significantly (t = 7.29, df = 1, p-value = 0.09). In the BGS traps, a total of 7,250 (63.4%) mosquitoes were collected. *Culex quinquefasciatus* (n = 3395; 46.8%) was the most abundant followed by *Ae. vittatus* (n = 1,551; 21.4%) and *Ae. aegypti* (n = 1,509; 20.8%). The primary vectors of dengue and chikungunya in the region were distributed across the study areas as observed in the study (Table 1).

### Seasonal variation in *Ae. vittatus* abundance

Based on the mosquito collection, data from two seasons in June (long rainy season) and December (short rainy season) *Aedes vittatus* collections revealed a clear pattern of seasonal variation. In June, a total of 1,858 Ae. vittatus mosquitoes were captured across both sampling methods, 941 by Bioagent Sentinel (BGS) traps and 917 through Human Landing Collection (HLC). In contrast, the December collection yielded only 1,131 individuals 610 via BGS and 521 via HLC (Table 2). This represents an approximate 39% overall decline in abundance between the two sampling seasons. The observed decline was consistent across both collection methods, with the BGS trap showing a 35% decrease and the HLC method showing a 43% decrease from June to December.

Although the BGS traps were deployed continuously for 24 hours (day and night), they captured a slightly higher overall number of mosquitoes compared to the Human Landing Catch method, which was conducted over a shorter 6-hour period. However, this difference in total mosquito catch between the two methods was not statistically significant, as indicated by the Kruskal-Wallis test ($\chi^2 = 2.65$, df = 1, p = 0.1). Further analysis of mosquito abundance across individual collection sites revealed notable variations in the relative performance of the two sampling methods. At the Mama Ngina Drive site, the HLC method outperformed the BGS traps, yielding a higher number of mosquitoes. In contrast, at the Railway Station site, BGS traps collected more mosquitoes than the HLC method, indicating better performance of the traps at this

**Table 1. Mosquito species sampled using the Human landing collection and Biogent sentinel traps in Mombasa city in 2021.**

| Species | Biogent Sentinel Trap(BG) | | Human Landing Collection(HLC) | | Total collection | |
| --- | --- | --- | --- | --- | --- | --- |
| | BG Day | BG Night | Morning | Afternoon | Total collected | Percentage (%) |
| *Ae. aegypti* | 265 | 1244 | 956 | 1260 | 3725 | 32.6 |
| *Ae. vittatus* | 12 | 1539 | 665 | 773 | 2989 | 26.1 |
| *Cx. quinquefasciatus* | 4 | 3391 | 71 | 172 | 3638 | 31.8 |
| *Ae. simpsoni* | 13 | 0 | 32 | 44 | 89 | 0.8 |
| *Cx. zombaensis* | 0 | 373 | 49 | 68 | 490 | 4.3 |
| Others species | 45 | 364 | 58 | 37 | 504 | 4.4 |
| **Total** | **339** | **6911** | **1831** | **2354** | **11435** | **100** |

**Table 2. Seasonal Variation in mosquito species sampled using the Human landing collection and Biogent sentinel traps in Mombasa city.**

| Species | Biogent Sentinel Trap (BG) | | Human Landing Collection (HLC) | | Total collection | |
|---|---|---|---|---|---|---|
| | June | December | June | December | Total collected | Percentage (%) |
| *Ae. aegypti* | 859 | 650 | 865 | 1351 | 3725 | 32.6 |
| *Ae. vittatus* | 941 | 610 | 917 | 521 | 2989 | 26.1 |
| *Cx. quinquefasciatus* | 2141 | 1254 | 240 | 3 | 3638 | 31.8 |
| *Ae. simpsoni* | 0 | 13 | 0 | 76 | 89 | 0.8 |
| *Cx. zombaensis* | 360 | 13 | 82 | 0 | 490 | 4.3 |
| Others species | 280 | 129 | 27 | 103 | 504 | 4.4 |
| **Total** | **4581** | **2669** | **2131** | **2054** | **11435** | **100** |

location. Interestingly, at the Kenya Ports Authority site, both sampling methods were equally effective, each recording an identical number of mosquitoes.

These site-specific differences in mosquito capture rates suggest that the effectiveness of mosquito collection methods can vary considerably depending on local environmental or ecological conditions. Factors such as habitat type, mosquito species composition, human activity, and microclimatic variations may influence trap performance and mosquito behavior, thereby affecting the efficiency of sampling techniques (Fig 2).

## Relative abundance

Given the skewness of the mosquito species abundance and distribution, we also assessed the relative abundance of each species across collection methods and sites. Relative abundance was calculated as the proportion of each species count relative to the total count per species. Our results showed that *Ae. aegypti* dominated mosquito communities across all sites and collection methods, consistent with its established role as the primary urban vector of CHIKV and DENV in the region. However, the invasive *Ae. vittatus* was also notably present, particularly in BG-Sentinel trap collections across all sites (S1 File). While *Ae. aegypti* was more frequently captured by HLC, *Ae. vittatus* consistently appeared in both HLC and BG collections. BG-Sentinel traps captured a more diverse mosquito community, including several less abundant species such as *Cx. vansomereni* and *Ae. simpsoni*, whereas HLC collections were primarily dominated by *Ae. aegypti* and *Ae. vittatus* (Fig 3).

## Alpha and beta diversity

We also calculated diversity metrics for the mosquito population sampled. To account for differences in sampling depth, mosquito counts were normalized using Scaling with Ranked Subsampling (SRS)**.** This preserves community structure by proportionally scaling species counts to a uniform depth, thereby avoiding the data loss and randomness introduced by rarefaction. Diversity metrics were calculated using the SRS-normalized abundance table. To determine an appropriate minimum sampling depth (Cmin) for SRS normalization, we first generated rarefaction curves using a step size of 50 mosquitoes. Based on the point at which the curves plateaued, a Cmin of 50 mosquitoes was selected, balancing species richness coverage with sample retention for downstream diversity analyses (Fig 4).

We assessed alpha diversity using species richness, Shannon index, and Simpson index for mosquito collections from BG-Sentinel traps and HLC, with six collectionsample for each method. While the BG-Sentinel trap yielded marginally higher mean values for species richness, Shannon diversity, and Simpson diversity, these differences were not statistically significant (Richness: $t = 0.63$, $df = 10$, $p = 0.54$; Shannon: $t = 0.32$, $df = 7.62$, $p = 0.75$; Simpson: $t = 0.08$, $df = 7.63$, $p = 0.94$) (Fig 5).

The model tested whether mosquito communities differed significantly between BGS traps and HLC methods. Ordination was visualized using Principal Coordinates Analysis (PCoA), where each point represented an individual mosquito

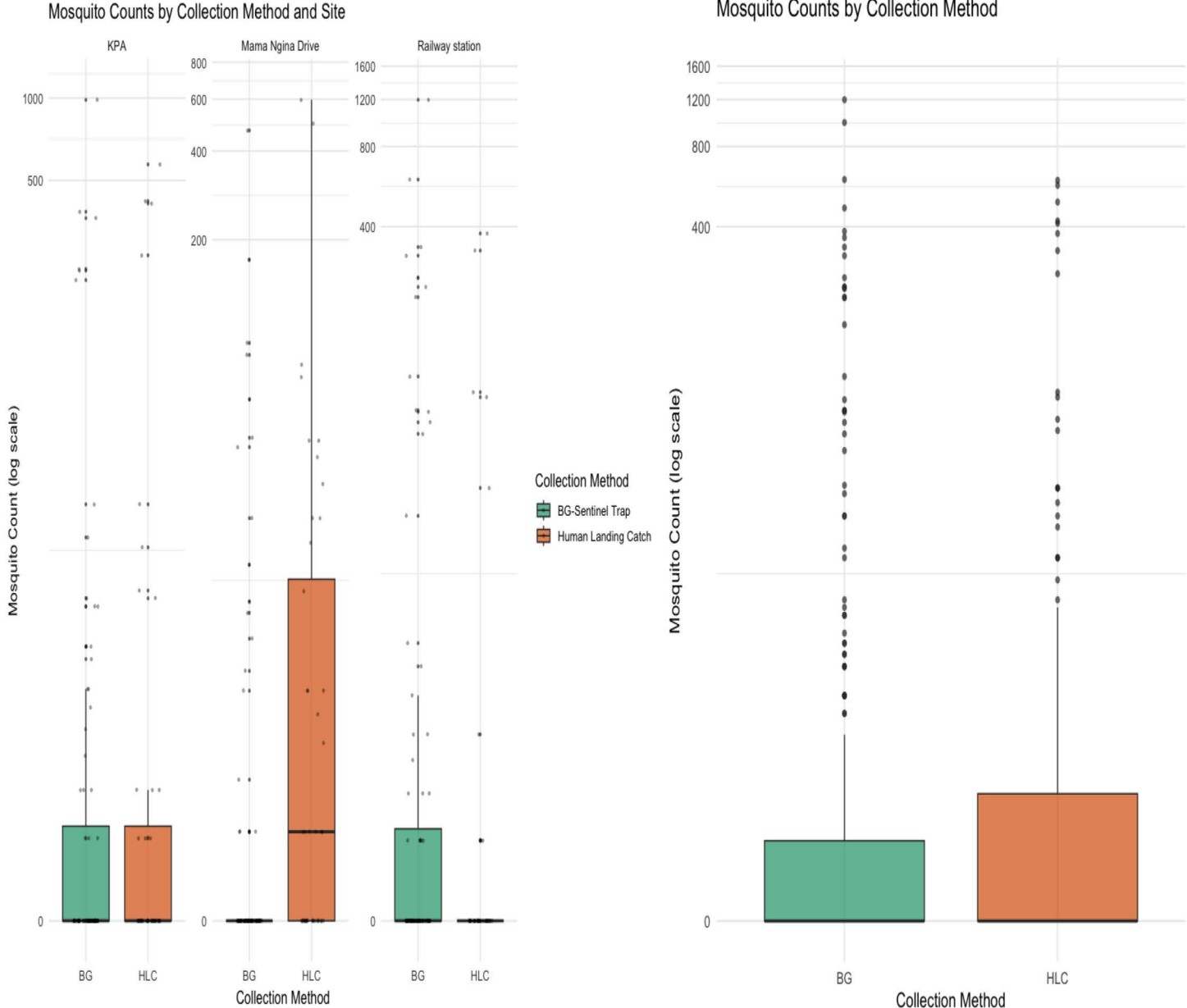

**Fig 2. Mosquito counts by collection method across the three sampling sites (Kenya Ports Authority, Mama Ngina Drive, and Mombasa Railway Station) in Mombasa, Kenya.** Boxplots showing the distribution of mosquito counts captured by BGS traps and HLC methods across the three sites. Each point represents the mosquito count from a single Mosquito sample, and boxes indicate the interquartile range (IQR) with medians. The y-axis is on a log scale to account for over dispersion in the data. While HLC collections showed slightly higher median counts, overall differences between methods were not statistically significant (Kruskal-Wallis test, p = 0.1).

collection method, and ellipses indicated 95% confidence intervals around group centroids. Our results showed no statistically significant differences in mosquito community composition between the two collection methods (PERMANOVA, R² = 0.15, F = 1.78, p = 0.16) (Fig 6).

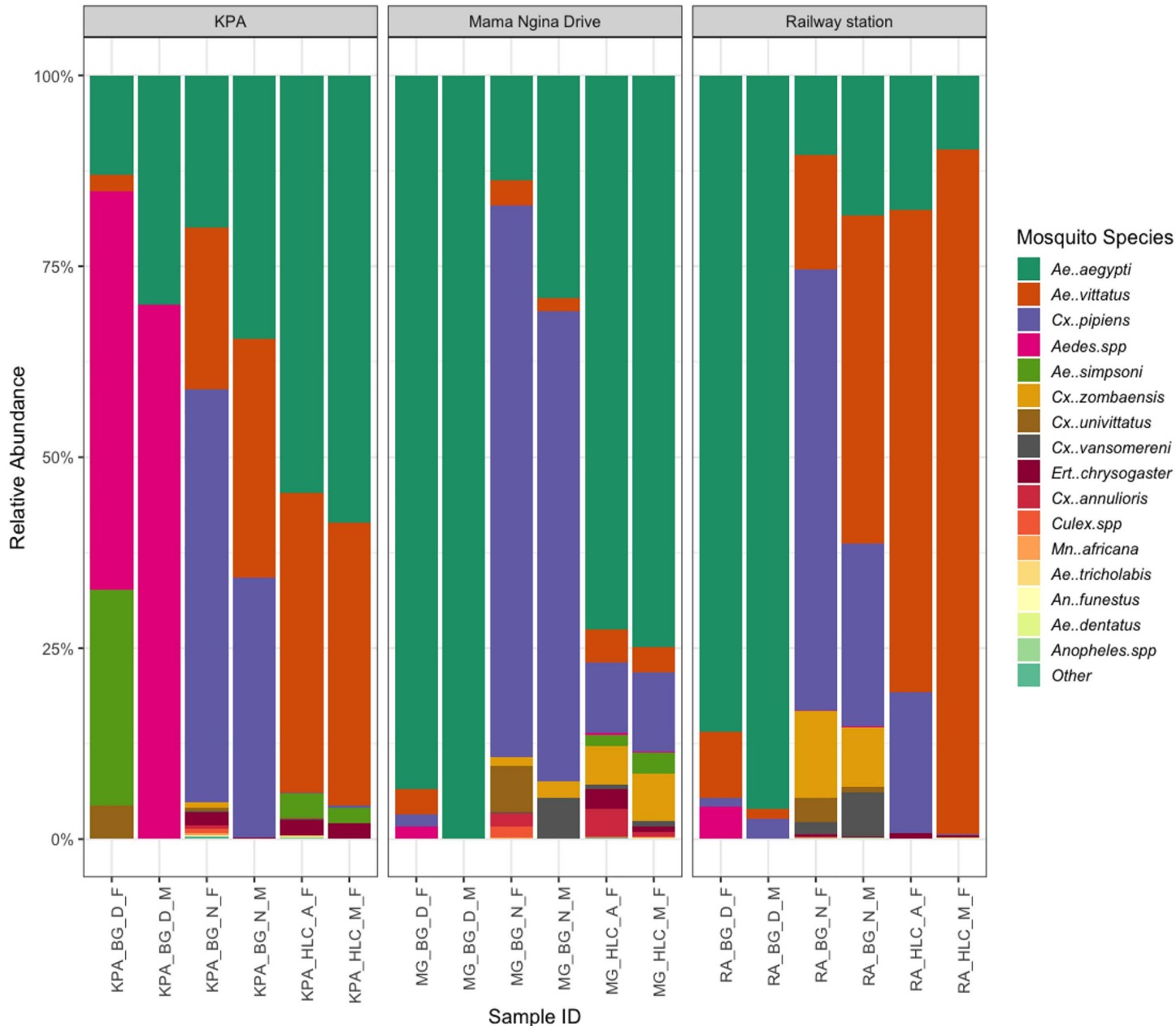

**Fig 3. Relative abundance of mosquito species by collection method across sites in Mombasa, Kenya Stacked bar plots show the relative abundance of mosquito species captured using BG-Sentinel traps and HLC methods across three sites: Kenya Ports Authority, Mama Ngina Drive, and Mombasa Railway Station.** Each bar represents a single collection period, with species proportions displayed as percentages. The y-axis indicates relative abundance, while the x-axis shows individual sample identifiers, which are named based on site (KPA = Kenya Ports Authority, MG = Mama Ngina Drive, RA = Mombasa Railway Station), collection method (BG or HLC), time of collection (D = day, N = night, A = afternoon, M = morning), and sex (M = male, F = female).

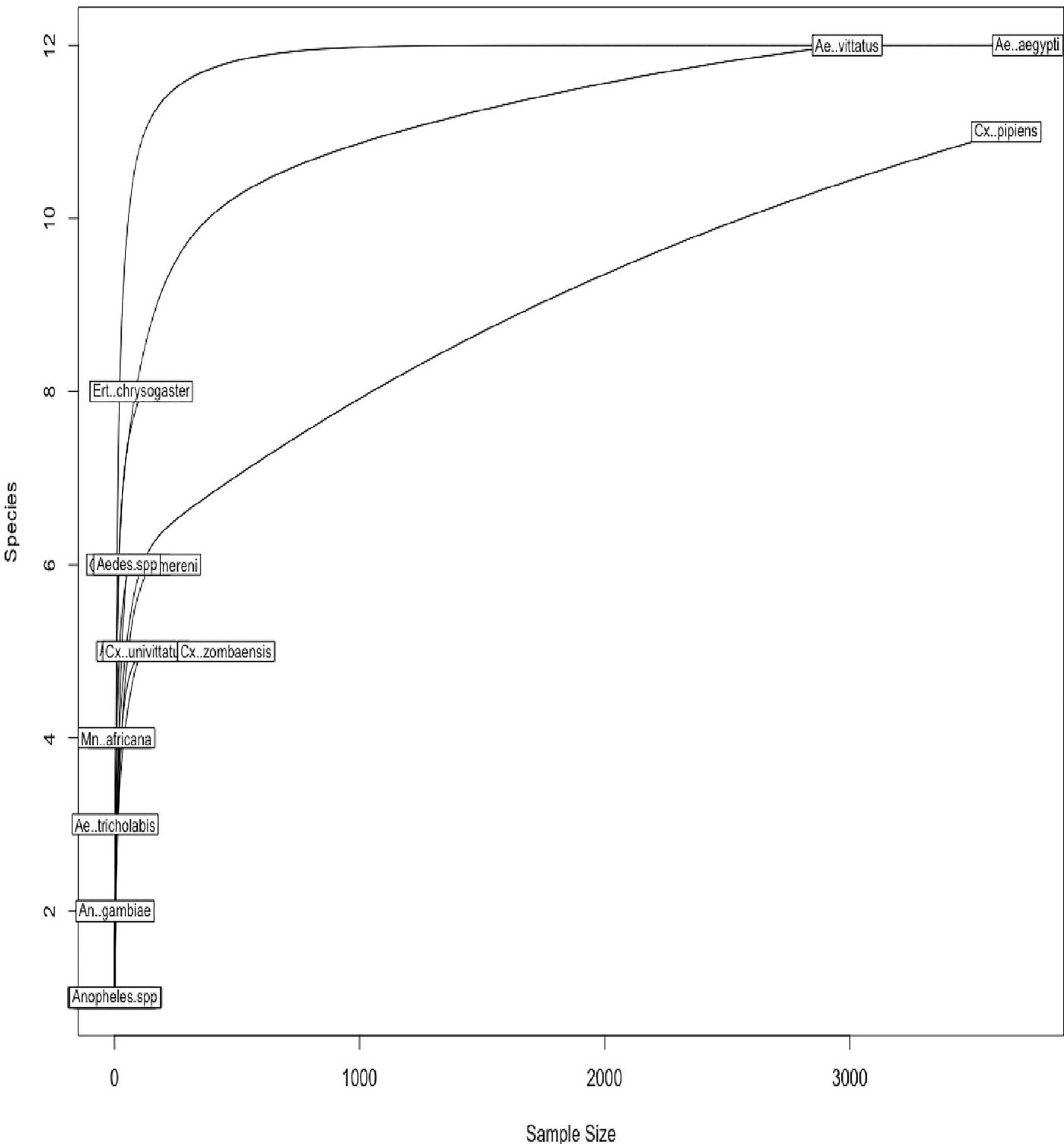

**Fig 4. Rarefaction curves of mosquito species richness by sampling depth.** Rarefaction curves showing the relationship between sampling depth (x-axis, number of mosquitoes) and observed species richness across individual samples. Each line represents a single mosquito species, illustrating how species accumulation plateaus with increased sampling effort. The majority of curves plateaued around 50 mosquitoes, indicating that this depth captures the most significant mosquito species present in most sampling period.

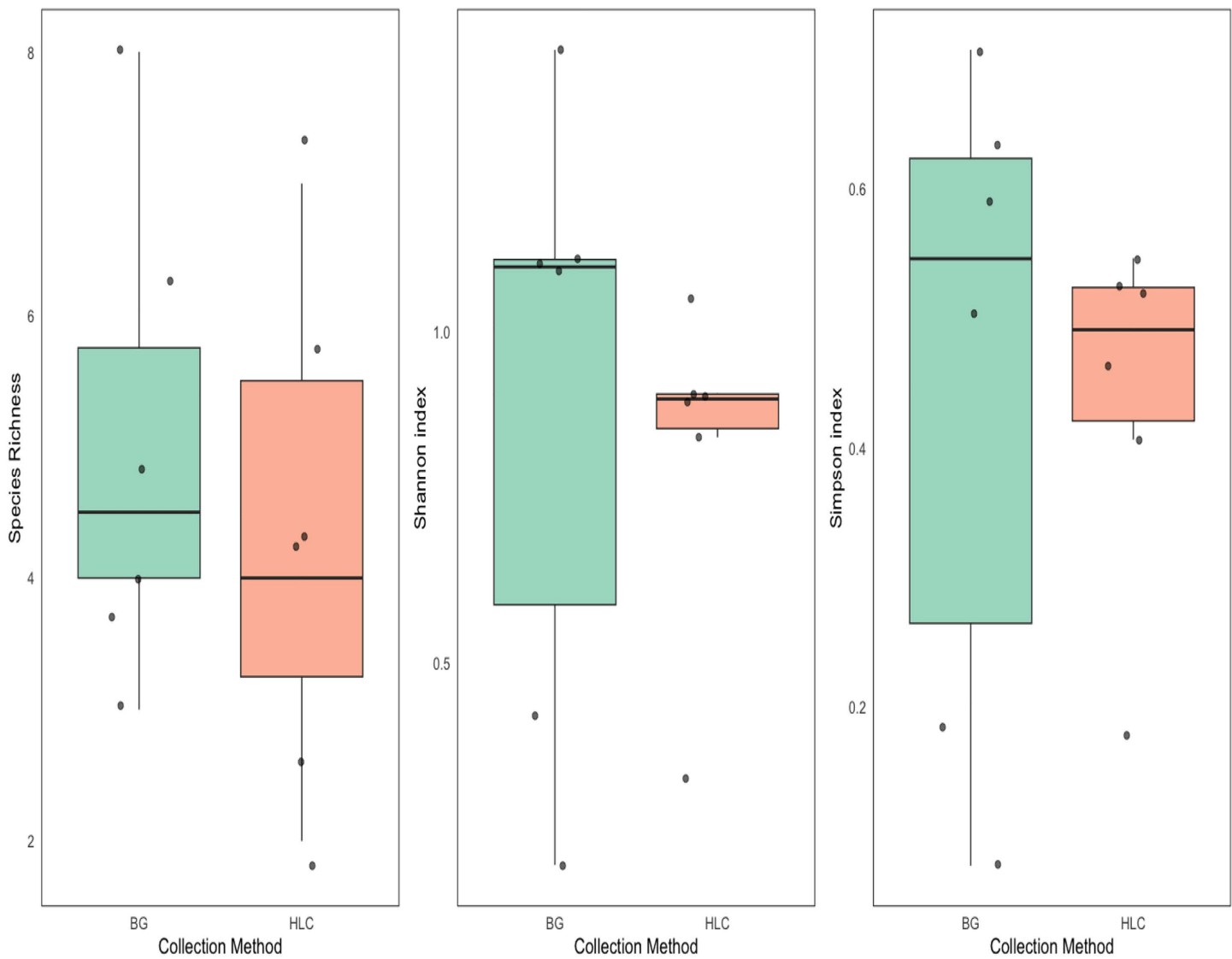

**Fig 5. Alpha diversity of mosquito species by collection method.** Boxplots show the distribution of species richness, Shannon index, and Simpson index across mosquito samples collected using BGS traps (green) and HLC (orange). Each dot represents an individual mosquito collection method. While mean diversity values appeared slightly higher in BGS samples, no statistically significant differences were observed between collection methods (t-test, p > 0.05 for all metrics).

### Molecular identification *Ae. vittatus* species

A total of 20 *Ae. vittatus* PCR amplicons were successfully sequenced, after trimming 9 haplotypes were obtained, Nucleotide BLAST (BLASTn) analysis of the cox1 sequences revealed 99–100% similarity with corresponding *Ae. vittatus* sequences available in the GenBank database, distinct from *Ae. aegypti* and *Ae. albopictus* confirming species identity with high confidence (Fig 7).

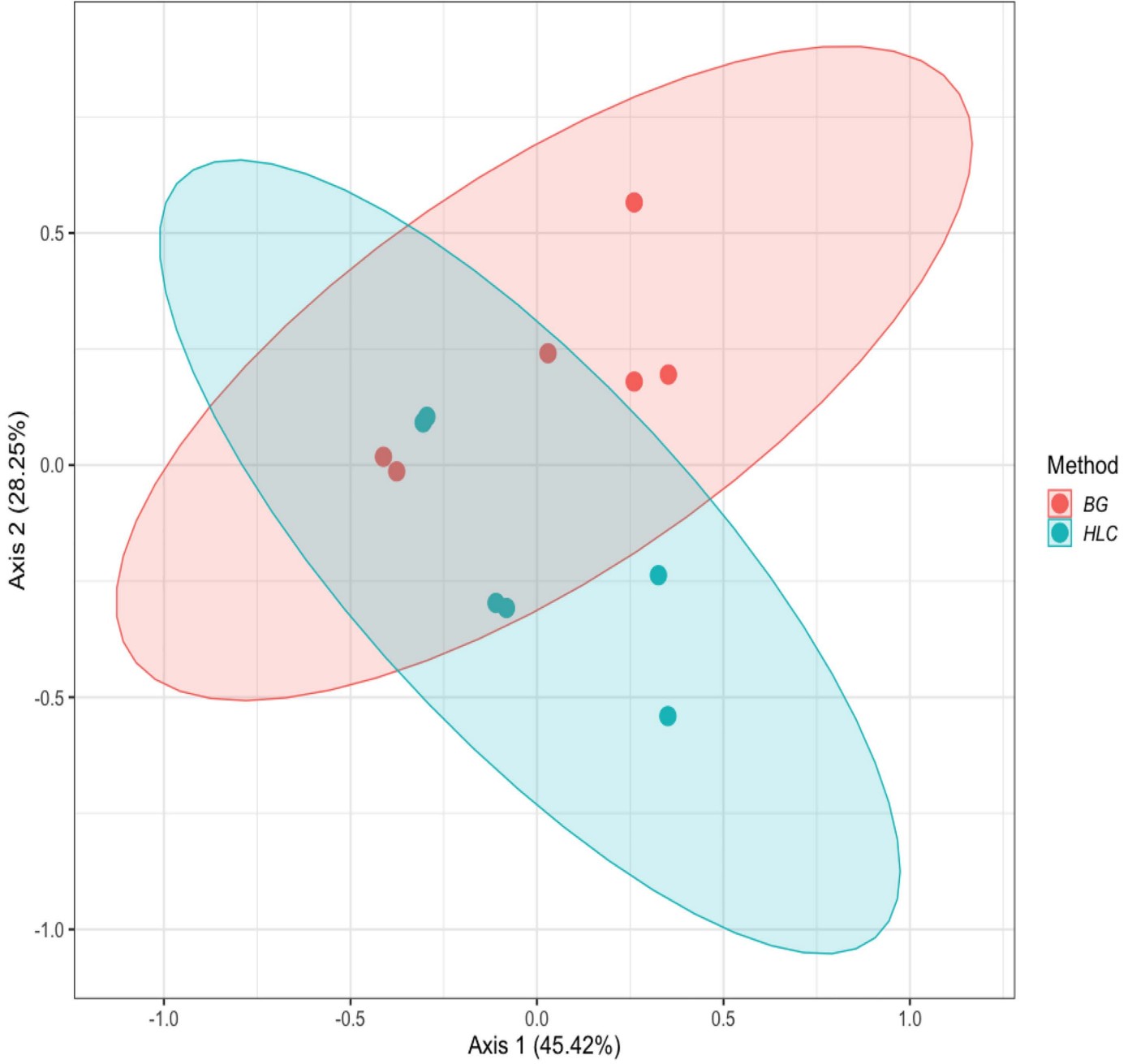

**Fig 6. Beta diversity of mosquito communities based on collection methods.** Ordination plot (PCoA/NMDS) of Bray-Curtis dissimilarities showing mosquito population composition across samples collected using BG-Sentinel traps (red) and HLC (blue). Each dot represents an individual mosquito collection period. While some clustering by collection method is observed, the substantial overlap of ellipses suggests limited differentiation in species composition between methods.

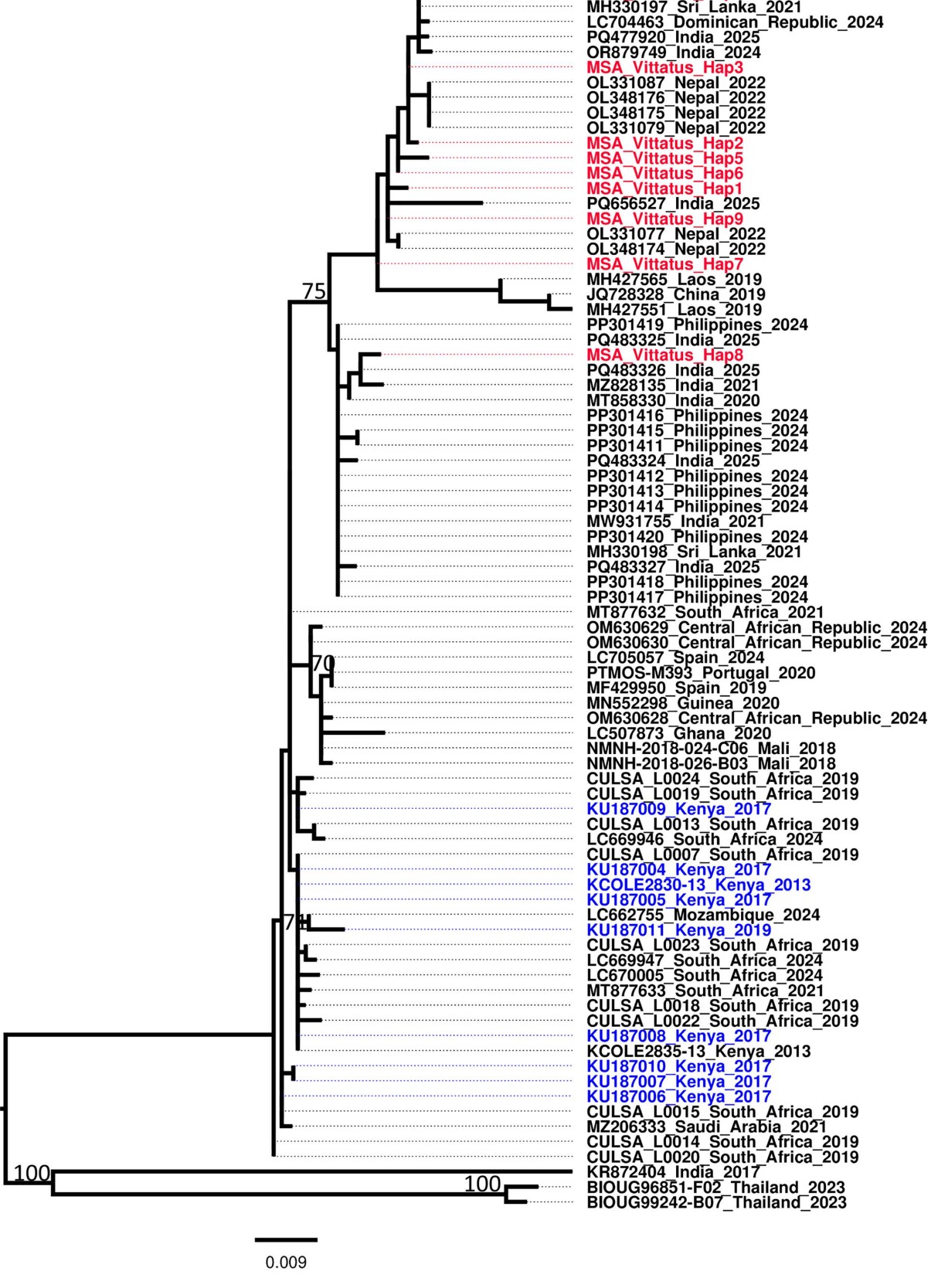

**Fig 7. Maximum likehood phylogeny of mitochondrial cox1 sequences of *Ae. vittatus* from Mombasa Haplotypes generated in the present study are highlighted in red, the previous Kenyan sequences from Kenya sequenced in 2017 highlighted in blue while references sequences from GenBank are represented by their accession number, country of origin and species name.** The site in our study is abbreviates in three letter words (MSA = Mombasa).

## Mosquito biting activity

Mosquito catches from HLC were used to characterize the biting activity of female *Aedes* mosquito species. Variation in diurnal biting behavior was assessed by comparing catches from morning (09:00–12:00) and afternoon (14:00–17:00) sampling periods. The biting rate of *Ae. vittatus* was 1.0 bite/person/hour in the morning and 1.1 bites/person/hour in the afternoon. Statistical analysis indicated no significant difference in biting rates across the two time periods ($t = 5.57$, $df = 1$, p-value = 0.11). Similarly, *Ae. aegypti* exhibited high biting activity during both periods, with an average of 1.4 bites/person/hour in the morning and 1.8 bites/person/hour in the afternoon; however, this difference was not statistically significant ($t = 4.17$, $df = 1$, p-value = 0.15) (Fig 8A). Biting activity patterns for both *Ae. aegypti* and *Ae. vittatus* were consistent across all study sites, with biting activity during morning and afternoon hours being uniform (Fig 8B). Also, there was no difference in biting rates between *Ae. aegypti* and *Ae. vittatus* ($t = 6.0081$, $df = 1$, p-value = 0.105). In contrast, *Cx. quinquefasciatus*, a primarily nocturnal vector, showed a much lower biting rate of 0.4 bites/person/hour during daytime collections, which is expected since this species is a known late evening feeder. *Aedes simpsoni* which is also a competent vector of chikungunya exhibited biting rates of 0.2 b/p/h.

## Chikungunya genotype

Among the 842 mosquito pools tested, one pool of *Ae. vittatus* sampled from Mombasa railway station in 2021 exhibited a clear reproducible cytopathic effect in Vero cell lines. Metagenomics next-generation sequencing (mNGS) recovered near-complete sequence of CHIKV (99.2%), with a high level of support of approximately > 13874x coverage. The consensus genome was compared to publicly available sequences in the Genbank nt/nr database using the BLASTn tool, and it was found to be closely related to strains that have been associated with outbreaks in this region. Phylogenetic analysis placed the CHIKV strain (MSA-1200) in the ECSA genotype (Fig 7). Similar to BLAST analysis, the strain clustered together with previous CHIKV strains isolated from Kenya mostly during 2018 outbreak [3], as well as other circulating strains in the region especially from the horn of Africa (Fig 9).

## Discussion

Globally, *Ae. aegypti* and *Ae. albopictus* are well-known vectors of viruses including chikungunya and dengue. In Kenya, where *Ae. albopictus* has not been documented, *Ae. aegypti* remains the principal vector of dengue and chikungunya viruses especially in Mombasa city where these diseases are endemic. Studies conducted in coastal regions have identified various breeding habitats of *Ae. vittatus* alongside *Ae. aegypti* in containers such as tyres, water tanks, domestic containers, indicating a shift in vector behavior likely driven by urbanization [21]. *Aedes Aegypti* is a principle vector of Chikungunya however, the invasion of the traditionally sylvatic, and highly competent *Ae. vittatus* species raises questions as to its role in the transmission of these viruses. Additionally, the presence of *Ae. simpsoni*, although at low prevalence, which is a vector of CHIKV, and *Cx. quinquefasciatus* which is confirmed to transmit the virus, further enhances the risk of transmission while complicating disease control efforts. The increasing establishment of the predominantly sylvatic *Ae. vittatus* in Mombasa city, where they were previously absent [20], demonstrates its adaptability to urban environments and changing vector bionomics [13] as well as interaction with human populations. Our findings are also in agreement with recent reports of significant numbers in Mombasa [4]. Studies in other African countries have also reported *Ae. vittatus* in their urban settings [37,38].

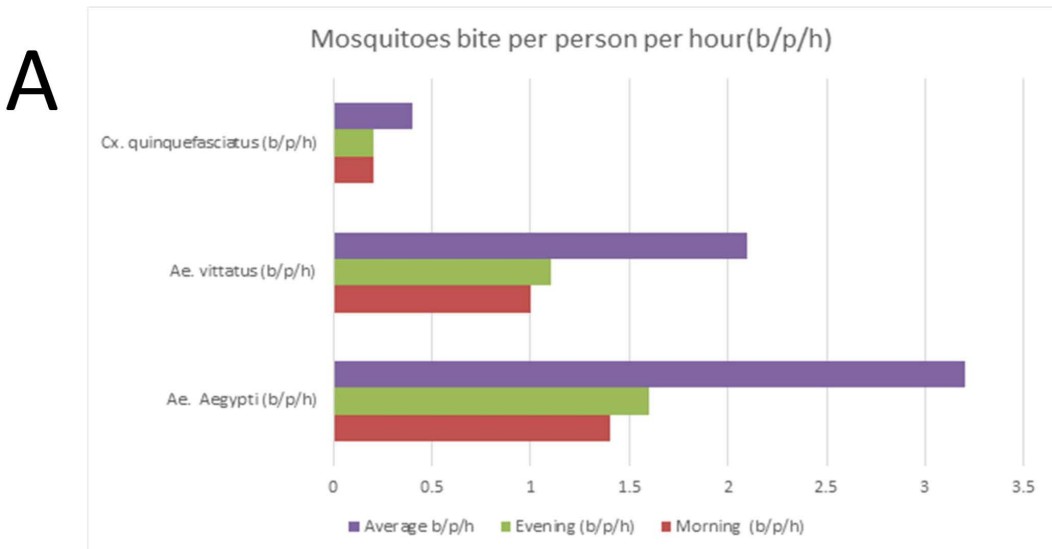

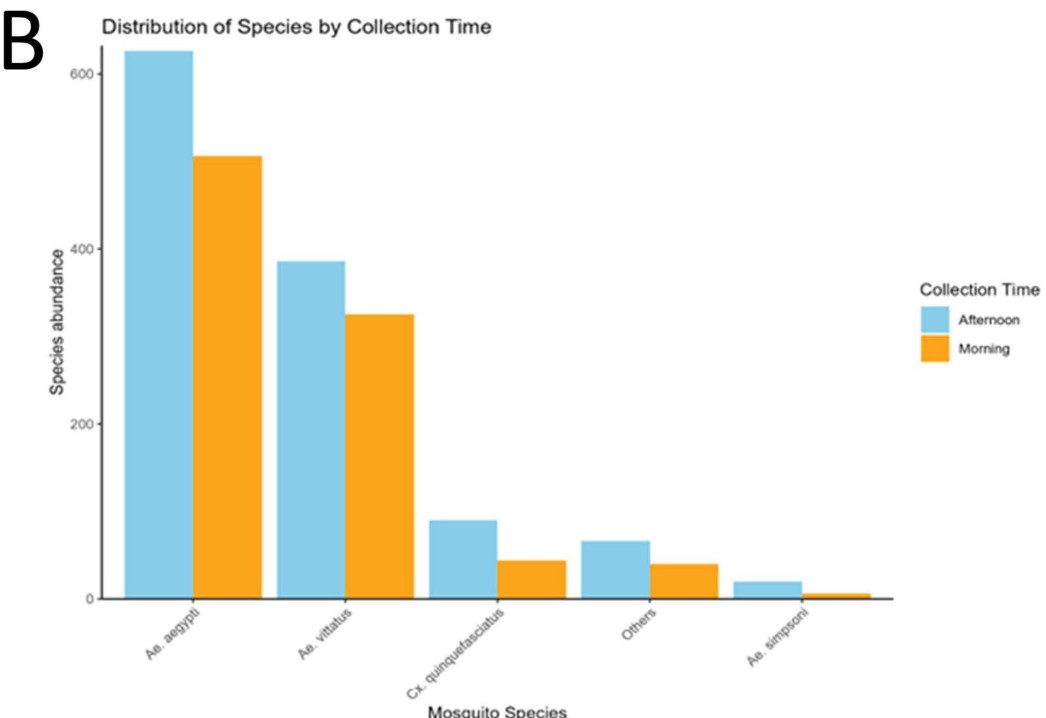

**Fig 8. The mosquito's bites per person per hour (b/p/h) (A) and abundance of vectors of CHIKV (B) during the morning and afternoon and hourly activity.**

In this study, both morphological identification and molecular verification were employed to confirm *Ae. vittatus* species. This is particularly important due to the frequent misidentification of specimens as *Ae. cogilli* in GenBank, as noted in previous studies [39]. Phylogenetic analysis using a > 600 bp segment of the mitochondrial COI gene confirmed that the specimens are closely related to populations found near Lake Victoria in western Kenya [40]. This finding highlights the

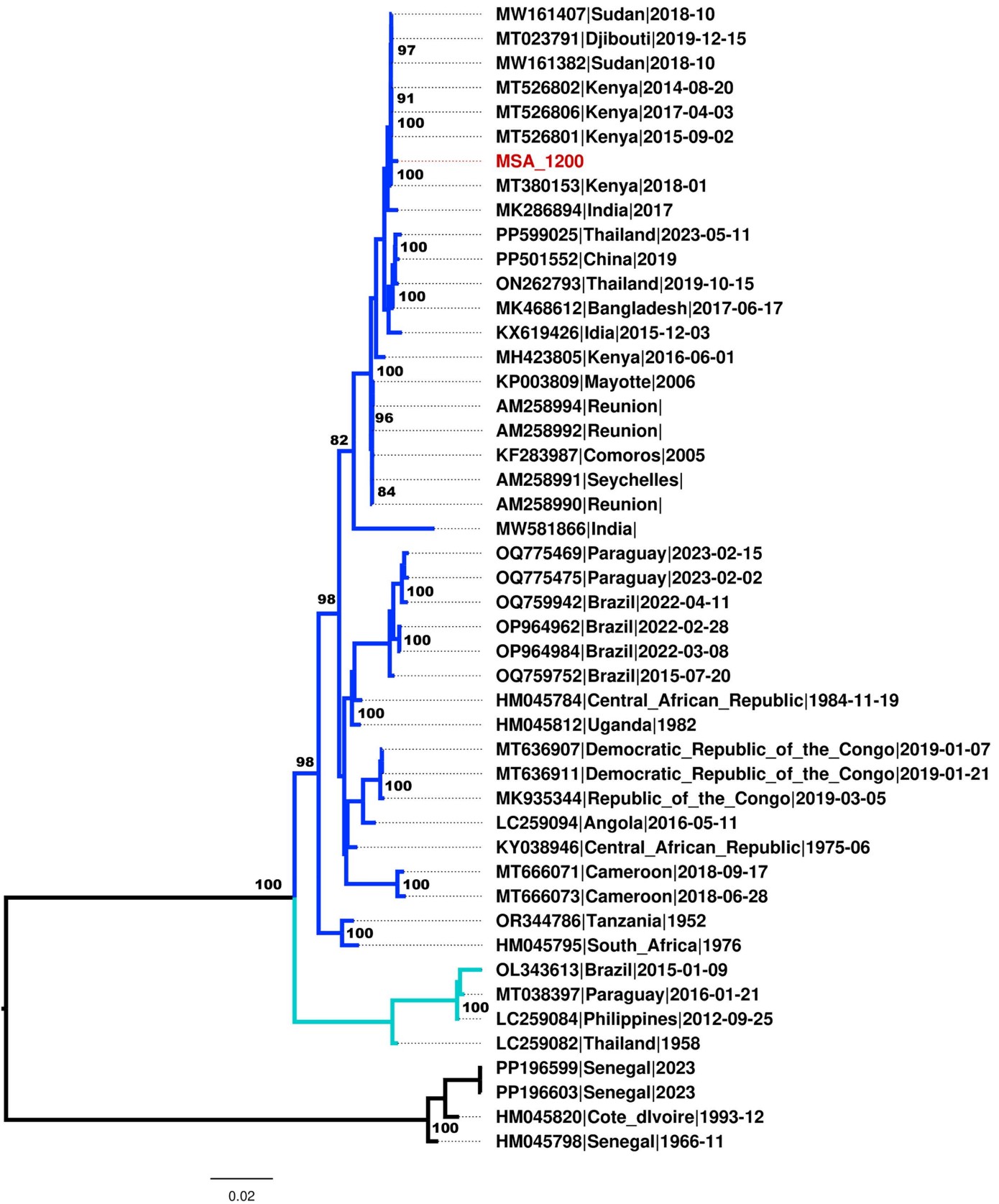

**Fig 9. Maximum likelihood phylogenetic tree of the CHIKV strain isolated from *Ae. vittatus* from Mombasa in 2021.** Mombasa strains, which are distinguished by a red highlight, clustered with strains isolated from Kenya mostly during 2018 outbreak. Percent bootstrap support is indicated by the values at each node (the values < 80 are omitted). The blue colour represents the ECSA genotype, light green colour the Asian and Caribbean genotype and the black represents the West Africa genotype.

potential role of increased trade in facilitating the introduction of invasive mosquito species. Further research is necessary to fully understand the origins and dispersal patterns of this mosquito species throughout the country. While the mode of introduction remains unknown, it could be due to human encroachment into the natural environment, and subsequent destruction of the preferred breeding habitats, due to rapid urbanization of the county. Previous studies have shown that the vector can transmit CHIKV [12,22]. This, together with isolation of CHIKV from the species, suggests that the *Ae. vittatus* may play an expanding role in disease transmission dynamics in its established urban habitat.

Our study has demonstrated that, just like *Ae. aegypti*, *Ae. vittatus* exhibits higher biting rates during morning and afternoon hours, suggesting that both species have similar biting periods between 09:00 am- 12:00pm, and14:00 pm and 17:00pm. These peak periods coincide with the active outdoor working hours of residents which could expose them to multiple infectious bites thus increasing the risk of infection. Previous studies also reported similar two-peak biting activities [17,41]. The human biting rates of *Ae. vittaus* were similar to those of *Ae. aegypti* showing the two vectors have got high public health importance in the area. Thus, Mombasa region has a high risk of arbovirus transmission based on the human vector counts that exceeded two female bites per hour which is indicative of a significant risk of virus transmission [42,43].

Understanding the seasonal dynamics of mosquito populations is crucial for assessing vector-borne disease risk and optimizing control strategies. From the study *Ae. vittatus* had high catches during the long rainy season compared to short rainy season. The reduction in vector numbers is likely driven by seasonal environmental factors rather than sampling bias. Potential influencing factors may include variations in rainfall, temperature, breeding site availability, and host-seeking behavior. From an epidemiological standpoint, this seasonal fluctuation is significant, *Ae. vittatus* serves as a competent vector of arboviruses such as dengue, chikungunya, or Zika, its higher activity during the June period could correspond with an increased risk of disease transmission. The alignment of peak mosquito activity with certain climatic conditions underscores the importance of timing vector surveillance and control efforts to coincide with periods of heightened mosquito abundance. This seasonal pattern reinforces the need for proactive vector monitoring and targeted interventions ahead of peak mosquito seasons. Incorporating such temporal insights into vector control programs can enhance the efficiency of resource allocation and reduce the public health impact of mosquito-borne diseases.

The collection of *Ae. vittatus* and *Ae. aegypti* at night by BGS traps points to opportunistic feeding behaviour where the Kenyan species could be adapting to nocturnal feeding in addition to their characteristic diurnal feeding behavior [17,44]. However, the exact time of the night when *Ae. aegypti* actively seeks hosts for feeding remains unknown. Previous studies have also reported nocturnal and diurnal activities for *Ae. vittatus* in rural areas, which may increase risk of disease transmission [13,45]. This is similar in our study where the vector was collected during the night by the BG traps indicating that in the region the vector has got both nocturnal and diurnal activities.

Urban environmental conditions and vector bionomics can influence biting times due to factors like artificial lighting and temperature variations. For instance, well-lit areas may extend mosquito activity into the night, altering traditional biting patterns [46]. Correlating mosquito abundance from BGS traps with human biting rates from HLC can improve estimates of disease transmission risk. A strong positive correlation would validate BGS traps as a reliable proxy to HLC despite being the gold standard for measuring human-vector contact, especially in settings where HLC is ethically or logistically challenging. This would enhance routine surveillance, support targeted vector control, and promote more ethical, scalable monitoring strategies. From an epidemiological perspective, such a correlation enhances the utility of BGS traps for routine surveillance and risk assessment. If higher trap catches consistently align with elevated biting rates on humans,

public health officials can use trap data to identify hotspots of transmission risk, monitor intervention effectiveness, and guide vector control strategies more efficiently.

Mombasa city has recorded an upsurge of chikungunya outbreaks in the recent past. Isolation of CHIKV from a pool of field collected *Ae. vittatus* species highlights the potential role of *Ae. vittatus* in these outbreaks, and the natural maintenance of the virus in nature. The virus strain is closely related to the Kenyan outbreak strains which were detected in 2015 and 2018, suggesting prolonged circulation in the coastal region [3]. The strain also clustered with those associated with chikungunya epidemics in South Asia, notably India, Somalia and Djibouti suggesting the movement of the ECSA strains across regions and continents, possibly facilitated by travel and trade. The increasing presence of the vector and the involvement of multiple mosquito species in CHIKV transmission in the Kenyan coastal region highlight the importance of integrated vector control strategies. A previously known bridge vector interlinking sylvatic with urban transmission cycle [47] due to its sylvatic and peri urban habitats, its establishment in cities would make it more involved in urban transmission of the virus.

The presence of two competent vectors capable of spreading the virus could potentially escalate outbreaks, highlighting the urgent need to enhance surveillance, and implement effective integrated vector control strategies. Moreover, public health education on personal protection measures, such as the use of insect repellents, is essential to reduce human-vector contact. A comprehensive understanding of the bionomics of vectors, including seasonal abundance and host-seeking behavior, is critical to predicting and managing future outbreaks.

Although the cross-sectional design of our study conducted across different seasons in the areas limits the ability to draw longitudinal conclusions, the findings offer valuable insights into the invasion of *Ae. vittatus* in Mombasa city. Another limitation is the absence of parity dissections or age-grading, which restricts our ability to determine whether the vector can survive long enough to sustain CHIKV transmission.

## Conclusion

The detections of chikungunya virus in *Ae. vittatus* mosquitoes in Mombasa underscores the complex nature of arbovirus transmission that is usually influenced by multiple mosquito species. It is evident that the recent high abundance of the vector in Mombasa has major implications for arbovirus transmission and control efforts in the region. Adaptability of *Ae. vittatus* and its competence as a vector highlights the need for comprehensive vector control strategies. Finally, diversity indices serve as important ecological indicators. Changes in diversity over time may signal shifts in species composition and abundance that could alter arbovirus transmission risk. The present results therefore provide a baseline for future surveillance and ecological monitoring in the region. In addition, these indices highlight the variety of mosquitoes present at the study site and their relative proportions, reinforcing our conclusion that comprehensive vector control strategies must target multiple species rather than a single dominant vector. Environmental and societal factors that influence the entomological factors and diseases transmission dynamics contributes to the spread of disease, emphasizing the importance of enhanced surveillance, targeted interventions, and strengthened public health preparedness.

## Supporting information

**S1 File. Distribution of the mosquito's species per site, collection period and collection time in Mombasa city.**
(XLSX)

## Acknowledgments

We acknowledge the technical support of Viral Hemorrhagic Fever Lab team members (KEMRI, Nairobi). We are grateful to Samuel Owaka, GIS support unit, KEMRI for generating the study map of the study area. We are also grateful for the support from the county staff, local chiefs as well as community members of Mombasa City.

## Author contributions

**Conceptualization:** Francis M. Musili, Rosemary Sang, Armanda Bastos, Joel Lutomiah.

**Data curation:** Francis M. Musili, James Mutisya, Solomon Langat, Betty Chelangat, Victor Anyango, Edith Chepkorir, Samson Konongoi, Rosemary Sang, Armanda Bastos, Joel Lutomiah.

**Formal analysis:** Francis M. Musili, Betty Chelangat, Victor Anyango, Edith Chepkorir, Samson Konongoi, Rosemary Sang, Armanda Bastos, Joel Lutomiah.

**Funding acquisition:** Rosemary Sang, Joel Lutomiah.

**Investigation:** Francis M. Musili, Rosemary Sang, Armanda Bastos, Joel Lutomiah.

**Methodology:** Francis M. Musili, James Mutisya, Solomon Langat, Betty Chelangat, Victor Anyango, Edith Chepkorir, Samson Konongoi, Rosemary Sang, Armanda Bastos, Joel Lutomiah.

**Project administration:** Joel Lutomiah.

**Resources:** Rosemary Sang, Joel Lutomiah.

**Supervision:** Rosemary Sang, Armanda Bastos, Joel Lutomiah.

**Validation:** Francis M. Musili, Rosemary Sang, Armanda Bastos, Joel Lutomiah.

**Visualization:** Francis M. Musili, Rosemary Sang, Armanda Bastos, Joel Lutomiah.

**Writing – original draft:** Francis M. Musili.

**Writing – review & editing:** Francis M. Musili, James Mutisya, Solomon Langat, Betty Chelangat, Victor Anyango, Edith Chepkorir, Samson Konongoi, Rosemary Sang, Armanda Bastos, Joel Lutomiah.

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
