## [Decision Letter · Decision Letter 0]

10 Sep 2025

PNTD-D-25-01359Role of native Aedes Aedimorphus vittatus in chikungunya virus transmission in the newly-invaded coastal island city of Mombasa, KenyaPLOS Neglected Tropical Diseases Dear Dr. Mulwa, Thank you for submitting your manuscript to PLOS Neglected Tropical Diseases. After careful consideration, we feel that it has merit but does not fully meet PLOS Neglected Tropical Diseases's publication criteria as it currently stands. Therefore, we invite you to submit a revised version of the manuscript that addresses the points raised during the review process. Please submit your revised manuscript within 30 days Nov 09 2025 11:59PM. If you will need more time than this to complete your revisions, please reply to this message or contact the journal office at plosntds@plos.org.  Please include the following items when submitting your revised manuscript:* A rebuttal letter that responds to each point raised by the editor and reviewer(s). You should upload this letter as a separate file labeled 'Response to Reviewers '. This file does not need to include responses to any formatting updates and technical items listed in the 'Journal Requirements' section below.* A marked-up copy of your manuscript that highlights changes made to the original version. You should upload this as a separate file labeled 'Revised Manuscript with Track Changes '.* An unmarked version of your revised paper without tracked changes. You should upload this as a separate file labeled 'Manuscript '. If you would like to make changes to your financial disclosure, competing interests statement, or data availability statement, please make these updates within the submission form at the time of resubmission. Guidelines for resubmitting your figure files are available below the reviewer comments at the end of this letter. We look forward to receiving your revised manuscript. Kind regards, Adly M.M. Abd-Alla, Prof asso.Section EditorPLOS Neglected Tropical Diseases Adly Abd-AllaSection EditorPLOS Neglected Tropical Diseases

Shaden Kamhawi

co-Editor-in-Chief

Paul Brindley

co-Editor-in-Chief

 **Journal Requirements:**

1) Please provide an Author Summary. This should appear in your manuscript between the Abstract (if applicable) and the Introduction, and should be 150-200 words long. The aim should be to make your findings accessible to a wide audience that includes both scientists and non-scientists. Sample summaries can be found on our website under Submission Guidelines:

2) We have noticed that you have uploaded Supporting Information files, but you have not included a list of legends. Please add a full list of legends for your Supporting Information files after the references list.

3) When completing the data availability statement of the submission form, you indicated that you will make your data available on acceptance. We strongly recommend all authors decide on a data sharing plan before acceptance, as the process can be lengthy and hold up publication timelines. Please note that, though access restrictions are acceptable now, your entire data will need to be made freely accessible if your manuscript is accepted for publication. This policy applies to all data except where public deposition would breach compliance with the protocol approved by your research ethics board. If you are unable to adhere to our open data policy, please kindly revise your statement to explain your reasoning and we will seek the editor's input on an exemption. Please be assured that, once you have provided your new statement, the assessment of your exemption will not hold up the peer review process.

 **Reviewers' comments:**  Reviewer's Responses to Questions

**Key Review Criteria Required for Acceptance?**

**Methods:**

-Are the objectives of the study clearly articulated with a clear testable hypothesis stated?

-Is the study design appropriate to address the stated objectives?

-Is the population clearly described and appropriate for the hypothesis being tested?

-Is the sample size sufficient to ensure adequate power to address the hypothesis being tested?

-Were correct statistical analysis used to support conclusions?

-Are there concerns about ethical or regulatory requirements being met?

Reviewer #1: Dear Authors,

The submitted manuscript is a result of impressive work that authors conducted in Mombasa. The results represent a significant contribution to the science and are applicable to the regular mosquito control strategies. From the epidemiological point of view obtained results are very useful in the fight against the diseases that are transmitted by identified vectors.

The language of the manuscript is understandable and should not be majorly revised.

However, there are many technical mistakes that have to be corrected before MS is published.

In the whole manuscript, but particularly in introduction, when the species is mentioned for the first time, the author and year of species description should be given.

Considering that the authors are talking about vector-borne diseases, the overview of the situation in Mombasa and Kenya should be given. Please give the exact numbers of human cases and epidemiological situation.

In the title, authors are focusing to Ae. vitatus, but they highlight the other vectors as well. The title has to be adapted to the story of the whole manuscript. Aedimorphus should be in bracket. Suggestion for the main title is: Role of mosquito vector species in chikungunya virus transmission and characterization of native Aedes vittatus in the newly-invaded coastal island city of Mombasa, Kenya

Please find my specific comments below:

L19 Missing space- December2021

L20 Did you mean Biogent instead of Bioagent?

L61 Alphavirus should be written with capital letter.

L84 Not clear. Please correct - area of Europe Africa, it is

L100 Not clear. Did you mean- giving its potential? Please revise the sentence. Same for Line 109.

L120 Please correct- (Figure (Fig) 1). to (Fig. 1) and delete comma before the bracket.

L143-144 This is not needed to be given in details - “while another copy was securely stored in a locked cabinet with restricted access in the PI’s office at KEMRI”. It should be shorten to- while another copy was kept in KEMRI.

L147 This sentence seems incomplete from the part- and compounds …

L147-157 Why did you adapt sampling to the biology of Ae. aegypti and you were sampling Ae. vittatus? You should explain that here or you should change the species in this section.

L154 Please correct - Lower part of legs.

L163 Missing space before bracket. And space in bracket should be deleted.

L176 Years should be deleted. Instead of (≤25 please write up to 25 or maximum 25.

L212 Why not in all 843 pools?

L264 Please give the citation for this formula.

L278 Instead- collected with, should be written – of which Ae. aegyptii is having….

L280 …than in the morning… Also space is missing before the bracket.

L281 Please give only two decimals in this line but the same is suggested for the whole manuscript (L192). Here should be – In the BGS

L286 Please delete- summary of

L294 No need to give the abbreviation of locations more than once in the text. Latter abbreviation or full name should be use, not both.

L298 Should be (Fig. 2).

L300 Title should not be in bold. Same for other figures.

L306 No need for italics.

L314 Please correct (Supplementary fig1). to (Supplementary 1).

L316 Missing space before the bracket. You should give the exact p value. It should be - significant difference or statistically significant difference.

L319 Should be (Fig. 3). Same in L339.

L331 Instead of community for mosquitoes please use the word population. Please correct accordingly in the whole manuscript.

L339 Missing full stop.

L347 Please delete subtitle because you already gave it above. Same for Beta subtitle afterward.

L385 Instead of mosquito collection event, it should be mosquito sample. Please correct in the whole manuscript.

L360 Give exact p value. If the p value is so low such as 0.00000005 in that case you should write p<0.001. If it is higher than 0.001 or the same as 0.001, you should write exact number. But also you should give exact statistical parameters for all. This version is not acceptable.

L382 Please fit the title and the figure on one page.

L385-386 The sentence in bold should be deleted and only left in bracket (MSA = Mombasa)

L400 Please correct to..... which is expected.... Or give a full stop before “that“.

Figure 8. Correct title. It is not human bite. It is mosquito bite. Please correct that in the whole manuscript. In A graph space is missing in title before bracket. A and B should be separated as two Figures.

L408 Missing space after year

L415 Citation should be given here.

L406 Please delete - identified in the study.

Fig 9 and its title should be fitted in the one page.

L427 vectors of viruses

L434 After CHIKV should be comma.

L439-440 Wrong format of citation.

L444 This is not correct to say that study confirmed that species is native because of that analyses. The analyses confirmed that the specimens which you analyzed are closely related to those that were before analyzed, also from Kenya.

L453 Missing full stop.

L462 Wrong format of citation.

L468 Delete space - areas ,

References:

What is - Kilcullen D. Mombasa: Gateway to Africa. 2019.? Please add link or more information.

Ref 22 should be corrected. It is WHO or write the full name. Please add the link.

The references require major revision. The space between the lines is not adequate, the name of the species is not in italics and there are technical mistakes such as missing spaces there that require corrections.

Reviewer #2: Yes the objectives are well stated and also the design of the study is good. My only concern is about the ethical approval of carrying out HLC (human landing) collection where Ae. aegypti is involved and there is no cure or medication for dengue. Anyway they have obtained clearance from their Ethics committee

Reviewer #3: (No Response)

**Results:**

-Does the analysis presented match the analysis plan?

-Are the results clearly and completely presented?

-Are the figures (Tables, Images) of sufficient quality for clarity?

Reviewer #1: Given in the text above.

Reviewer #2: The results have been well presented. On the whole it is good.

Reviewer #3: (No Response)

**Conclusions:**

-Are the conclusions supported by the data presented?

-Are the limitations of analysis clearly described?

-Do the authors discuss how these data can be helpful to advance our understanding of the topic under study?

-Is public health relevance addressed?

Reviewer #1: Given in the text above.

Reviewer #2: The paper is well written. My only concern is that there is no in depth discussion about what can be done to control the Ae. vittatus. Does Ae. vittatus breed along with Ae. aegypti in containers? Has it been found when larval surveys were carried out for Ae. aegypti by health teams. I know it is not part of this study but it can be included in the introduction or discussion.

The results shows that only one pool of Ae. vittatus was positive - thus it is an important vector. Even Ae. aegypti were not positive. Thus, for the benefit of the readers more information should be provided about Ae. vittatus.

Reviewer #3: (No Response)

**Editorial and Data Presentation Modifications?**

Reviewer #1: Given in the text above.

Reviewer #2: This is okay

Reviewer #3: (No Response)

**Summary and General Comments:**

Reviewer #1: Given in the text above.

Reviewer #2: As i have mentioned above, please provide more details about the breeding of Ae. vittatus. Also provide some methods that can be used by the surveillance team for the control. Most of my comments have been mentioned under the conclusion section.

Reviewer #3: General Comments

The manuscript by Mulwa et al. examines the abundance, biting activity, and potential role of Aedes vittatus in chikungunya virus (CHIKV) transmission in Mombasa, Kenya. The study is timely, relevant, and contributes valuable entomological and virological data, particularly the detection of CHIKV in Ae. vittatus. However, several interpretations are stronger than the evidence supports, and methodological details require clarification. See my comments below.

Major Comments

1. The high numbers of Ae. vittatus collected across multiple sites are suggestive of establishment in Mombasa. However, firm conclusions that the species has “adapted” and “successfully established” are not yet fully supported. No longitudinal data, larval surveys, or host-feeding studies are presented to confirm sustained presence. Moreover, the phylogenetic tree (Fig. 7) shows that the haplotypes from the present study (red) cluster distinctly from earlier Kenyan collections (blue), suggesting possible new introductions or reintroductions rather than continuity of a single established population. These claims should therefore be presented more cautiously, while acknowledging that the observed abundance provides preliminary evidence of establishment.

2. No parity dissections or age-grading were conducted, limiting assessment of whether Ae. vittatus can survive long enough to sustain CHIKV transmission. This should be acknowledged as a limitation.

3. CO₂ baited BG traps operated continuously for 24 h, while HLC was restricted to two 3-h daytime windows. Raw totals are therefore not directly comparable as presented in L290-298. Results should be standardized (e.g., mosquitoes per trap-hour vs. person-hour), and if possible, BG catches should be disaggregated by day vs. night. Otherwise, only within-method comparisons are defensible.

Also, an analysis correlating trap abundance (BG catches) with human biting rates (HLC) would strengthen epidemiological interpretation.

4. If collections in June and December could allow for seasonal comparisons, these results should be presented or explicitly noted as absent.

5. Figure 3 suggests that most of the Ae. vittatus captured using BG traps may also have been captured at night. This result is not highlighted in the results section.

6. How do the diversity indices sections (L330-367) add to the conclusions of this study?

7. Some aspects of the data presentation and analysis are unclear. For example, in Figure 8A the “average bite rate” appears higher than the rates reported from individual methods, which is difficult to interpret. Similarly, certain outputs (e.g., very low degrees of freedom) suggest results may be based on limited replication (i.e, analysis of total instead of replicate-level data). The authors should clarify their analytical approach for each comparison.

6. The opening paragraph of the Discussion section reads more like background. It should begin with a concise summary of the key findings (abundance, biting activity, CHIKV isolation) before contextualizing these results within the broader literature.

Minor comments

-Improve transitions between paragraphs in the introduction. Currently, paragraphs feel disjointed.

-Standardize “Biogent” (not “Bioagent”)

-Clarify the use of “native” versus “invasive” when describing Ae. vittatus. See full title and short title.

-State BG trap effort (trap-days) in the methods section.

-The HLC description (L147-165) emphasizes Ae. aegypti rather than Ae. vittatus. Clarify the rationale.

-Provide details on the study sites.

-Clarify which version of R was used for analysis.

-Table 1 should include site-level breakdowns.

-Avoid presenting the same data in text, tables, and figures.

-The phrase “multiple presence of the species” L38 is vague and should be clarified.

PLOS authors have the option to publish the peer review history of their article (what does this mean? ). If published, this will include your full peer review and any attached files.

**Do you want your identity to be public for this peer review?** For information about this choice, including consent withdrawal, please see our Privacy Policy .

Reviewer #1: No

Reviewer #2: No

Reviewer #3: No

  **Figure resubmission:**  While revising your submission, we strongly recommend that you use PLOS’s NAAS tool (https://ngplosjournals.pagemajik.ai/artanalysis) to test your figure files. NAAS can convert your figure files to the TIFF file type and meet basic requirements (such as print size, resolution), or provide you with a report on issues that do not meet our requirements and that NAAS cannot fix.

After uploading your figures to PLOS’s NAAS tool - https://ngplosjournals.pagemajik.ai/artanalysis, NAAS will process the files provided and display the results in the "Uploaded Files" section of the page as the processing is complete. If the uploaded figures meet our requirements (or NAAS is able to fix the files to meet our requirements), the figure will be marked as "fixed" above. If NAAS is unable to fix the files, a red "failed" label will appear above. When NAAS has confirmed that the figure files meet our requirements, please download the file via the download option, and include these NAAS processed figure files when submitting your revised manuscript. **Reproducibility:**  To enhance the reproducibility of your results, we recommend that authors of applicable studies deposit laboratory protocols in protocols.io, where a protocol can be assigned its own identifier (DOI) such that it can be cited independently in the future. Additionally, PLOS ONE offers an option to publish peer-reviewed clinical study protocols. Read more information on sharing protocols at https://plos.org/protocols?utm_medium=editorial-email&utm_source=authorletters&utm_campaign=protocols

---

## [Editor Report · Decision Letter 1]

10 Nov 2025

Dear Mr Mulwa,

We are pleased to inform you that your manuscript 'Role of invasive Aedes (Aedimorphus) vittatus in chikungunya virus transmission in the newly-invaded coastal island city of Mombasa, Kenya' has been provisionally accepted for publication in PLOS Neglected Tropical Diseases.

Best regards,

Adly M.M. Abd-Alla, Prof asso.

Section Editor

Adly Abd-Alla

Section Editor

Shaden Kamhawi

co-Editor-in-Chief

Paul Brindley

co-Editor-in-Chief

---

## [Editor Report · Acceptance letter]

Dear Mr Mulwa,

We are delighted to inform you that your manuscript, "Role of invasive Aedes (Aedimorphus) vittatus in chikungunya virus transmission in the newly-invaded coastal island city of Mombasa, Kenya," has been formally accepted for publication in PLOS Neglected Tropical Diseases.

Best regards,

Shaden Kamhawi

co-Editor-in-Chief

Paul Brindley

co-Editor-in-Chief
